# *Cor triatriatum dexter* in Dogs: A Silent Bystander or a Potential Time Bomb? A Focused Literature Review from a Professional Perspective

**DOI:** 10.3390/vetsci12020175

**Published:** 2025-02-14

**Authors:** Michela Pugliese, Diego Antonio Sicuso, Giordana Merola, Bengü Bilgiç, Annalisa Previti, Mehmet Erman Or, Annamaria Passantino

**Affiliations:** 1Department of Veterinary Sciences, University of Messina, 98168 Messina, Italy; diego150899@gmail.com (D.A.S.); giordana.merola@gmail.com (G.M.); annalisa.previti@yahoo.it (A.P.); annamaria.passantino@unime.it (A.P.); 2Faculty of Veterinary Medicine, İstanbul University-Cerrahpasa, 34098 Istanbul, Turkey; bengu.bilgic@iuc.edu.tr (B.B.); ermanor@iuc.edu.tr (M.E.O.)

**Keywords:** congenital heart diseases, dog, cardiac anomaly, atrial defect

## Abstract

*Cor triatriatum dexter* (CTD) is an uncommon congenital anomaly consequent to an abnormal separation of the atrium during embryological development, determining the persistence of the fibromuscular membrane dividing the right atrium. Based on the severity of the obstruction determined by the membrane, the clinical signs may be silent or appear as heart failure, with consequent ascites, and if there are other associated congenital cardiac anomalies, they could manifest as cyanosis or sudden death. The aim of the present review is to give a professional perspective, considering the defect from different points of view.

## 1. Introduction

Congenital heart diseases (CHDs) are structural anomalies present at birth, resulting from aberrant embryonic development. In veterinary medicine, CHDs are an important reason for disease and death in both dogs and cats [1]. These conditions can range from mild, clinically silent anomalies to severe defects that result in heart failure, cyanosis, or sudden death. The prevalence of CHDs varies across species and breeds, with certain defects being more common in specific breeds due to genetic predisposition [2,3,4].

*Cor triatriatum* (CT) is an uncommon congenital heart disease (CHD) described in humans, dogs, and cats. Humans, dogs, and cats present two different types, respectively, *Cor triatriatum dexter* (CTD) and *Cor triatriatum sinister* (CTS). CTD is characterized by the occurrence of an anomalous membrane dividing the right atrium into two compartments.

The most frequent locations of the abnormal membrane are at the right of the superior vena cava, coronary sinus, and inferior vena cava, and at the left of the coronary sinus and to the right of the superior and inferior vena cava [3]. This abnormality originates from the persistence of the embryonic Eustachian valve or the right sinus venosus during development. This structure typically exists in the embryonic stage within the right atrium to direct oxygen-rich blood from the cranial vena cava through the *foramen ovale* into the left atrium. In some cases, this structure does not regress properly, leading to an abnormal division of the right atrial chamber, which hinders normal venous return from the caudal vena cava and the coronary sinus [5].

The membrane dividing the atrium appears commonly perforated, more rarely it can be imperforate, determining a full obstruction of the venous flow through the caudal vena cava and/or from the coronary sinus [3,4].

In humans, CTD is an uncommon malformation appearing with various clinical signs. The first detection may occur at different ages. In children, its incidence is approximately 0.025% of CHDs. CTD may appear as an isolated defect or associated with other right-sided CHDs [6,7].

It can be associated with a range of other concurrent CHDs, such as tricuspid valve dysplasia [7,8,9,10], atrial septal defect [7], pulmonic stenosis [7,11], persistent *foramen ovale* [8,12,13], ventricular septal defect [7], persistent left cranial vena cava [14], and pulmonary and mitral valve dysplasia [7,10]. All of which may influence the prognosis [7].

The present review aims to (*i*) explore the pathophysiology of CTD and dogs; (*ii*) outline the clinical scenario underlying the increased interest in CHDs by overviewing the evolution of appropriate diagnostic techniques and the treatment options available; and (*iii*) provide a discussion from a medical–legal and ethical point of view.

### 1.1. Clinical Signs

The severity of clinical signs varies in relation to the dimension of membrane perforation, the position of the membrane in relation to the ways of right atrial inflow, the existence of atrial-level shunting, and any coexisting CHDs [7].

In adult people, different clinical signs including exercise intolerance, dyspnea associated with exercise, and supraventricular arrhythmias feeling like palpitations, syncope, and edema of limbs are reported [5,6]. Cyanosis is less frequently reported, and it is described in both adult and pediatric patients [5,15].

The age of the first appearance of clinical signs may be different. In fact, it may not appear until adulthood. They are imputable to right–left shunting through a concomitant presence of an atrial septal defect or patent *foramen ovale* [16,17,18,19,20].

In dogs, similarly to people, the age of detection may vary from puppyhood to senior dogs, but most cases report the occurrence of the first clinical signs as ascites in young patients [7,8,9,10,11,12,13,14,21,22,23,24,25,26,27,28,29,30,31]. For anatomic variables, determining interference with blood flow from the caudal vena and all concurrent CHDs associated with CTD may occur with different clinical signs [7].

Ascites and the consequential abdominal enlargement, respiratory distress, pleural effusion, exercise intolerance, stunted growth, and cyanosis are the most common findings [7]. Also, a cardiac murmur may be detected in a few patients [7]. Vomiting is reported as a sign correlated with abdominal effusion [7].

Table 1 summarizes the main clinical signs associated with CTD.

In Table 2 are summarized concurrent CHDs accompanying CTD in dogs.

### 1.2. Diagnosis

The diagnosis of CTD can be challenging due to the absence of clear clinical findings of right-sided heart failure, such as jugular evidence or pulses, craniomandibular or forelimb edema, and the lack of a heart murmur. To more accurately describe this complicated cardiovascular defect and to support in choosing the most proper treatment, diagnosis of CTD benefits from a multifaceted diagnostic method including various imaging tests (i.e., thoracic radiographs, echocardiography with doppler ultrasound and contrast-enhanced (“bubble study”), abdominal ultrasound, angiogram, computed tomography angiography, and a cardiac magnetic resonance) [7,10,14,21,27,28]. The echocardiogram is the gold standard for diagnosing CTD due to its ability to visualize the abnormal membrane, evaluate the dimension of the orifice, measure the turbulent flow among the high-pressure tail compartment and the low-pressure skull compartment, and assess associated congenital heart disease [5,6,7,26].

However, a complete echocardiography including two-dimensional, mono-dimensional, and Doppler evaluation plays an important role in identifying this condition by revealing specific features. Right parasternal four-chamber-long axis views and right parasternal oblique views focused on the right atrium and caudal vena cava allow the visualization of the abnormal membrane dividing the atrium [32]. The membrane dividing the atrium may have one or more orifices permitting blood flow between the atrial compartments or may appear imperforated. The size of these orifices may be evaluated from the right parasternal oblique view focused on the right atrium and caudal vena cava. It can vary considerably, and if they are very small or missing, they can cause significant obstruction to blood flow.

Doppler echocardiography is also useful for detecting turbulent blood flow through the orifices in the membrane, especially when using unconventional views. If the membrane is perforated, pulsed-wave or continuous-wave Doppler is applied to evaluate the pressure gradient (measured at peak velocity, usually at the end of ventricular systole in the absence of right ventricular pressure overload) between the atrial chambers [5,6,7]. Additionally, color Doppler aids in visualizing the blood flow between the chambers, highlighting pressure differences and abnormal flows caused by the membrane. Echocardiography can also assess the function of the atrial chambers, checking for dilation related to the flow obstruction. Moreover, CTD is often associated with other CHDs, such as atrial septal defects or valve abnormalities, which can also be identified and evaluated through echocardiography. Finally, in some cases, contrast echocardiography, known as a “bubble study”, can be used to enhance the visualization of the membrane and orifices, making the blood flow path between the chambers clearer [21]. Echocardiographic diagnosing of CTD often necessitates nonconventional views and color Doppler or contrast echocardiograms. Differentiating CTD from congenital caudal vena cava obstruction may be arduous in the absence of an angiographic and surgical evaluation [13,29]. Thoracic radiograms may also often appear normal even though different alterations may be detected. Usually, the presence of cardiomegaly (≥10.5; range: 10–12.25) is indicated by an increased vertebral heart score, suggesting the presence of cardiomegaly [33]. The most common abnormality detected on thoracic radiograms is a loss of cranial abdominal serosal detail in the observable part of the abdominal cavity. A distended dilated caudal vena cava is a less frequent sign. Also, dogs presenting with pleural effusion may show increased opacity within the pleural space [7].

Dogs affected by CTD usually show a sinus rhythm, associated with right axis deviation. In a few cases, P-pulmonale may be present, characterized by atrial enlargement. The occurrence of ventricular pre-excitation syndrome characterized by tachycardia occurring at rates up to 350 bpm is also documented [7].

### 1.3. Pharmacological and Surgical Treatment

The treatment approach depends on the presence of clinical symptoms strongly related to the anatomy of CTD, and the presence and size of communication between the two compartments. In the presence of large-size membrane communication, a diagnosis of CTD may be incidental, and there is no need to plan any therapeutic intervention [7]. In around 40% of dogs with CTD, the ostium presents as an appropriate size; therefore, therapeutic action is necessary to regulate the hemodynamic impact [7].

In more severe cases, pharmacological treatment only aims to control the clinical signs, given that definitive treatment is based on minimally invasive procedures or an invasive surgical approach [11,12,22,23,24,25,26,27,28,34,35,36,37,38]. Typical pharmacological treatment to control pre-surgery symptoms is based on the administration of diuretics such as furosemide and/or spironolactone, and/or ACE inhibitors, such as benazepril [7,13,29,35,39,40,41]. The use of diuretics helps to decrease fluid accumulation in the abdomen, reducing circulating blood volume and lowering venous pressure. This process alleviates the congestion typically associated with heart failure. In dogs, furosemide is often administered to relieve symptoms of heart failure, either orally or intravenously, depending on the severity of the condition [39]. Therefore, dosing must be carefully managed to prevent complications such as dehydration and electrolyte imbalances, especially hypokalemia [39].

Vasodilators, particularly ACE inhibitors such as enalapril or benazepril, help to treat heart failure and hypertension. Benazepril is preferred in patients with renal dysfunction because its metabolism is more balanced between the liver and kidneys, thus reducing its impact on the kidneys [31,32,33,34,35,36,37,38,39,40,41,42]. During post-surgery, low-dose aspirin could be used to reduce the risk of thrombus formation [41]. Instead, if symptoms restarted during the follow-up period, furosemide could be administrated to stabilize the patient before the next surgery [7].

Several therapeutic procedures are described for correcting CTD including standard balloon dilation, cutting balloon dilation [7,22,23,26,27], hybrid technique [43], atriotomy with direct removal of the membrane with inflow occlusion [9,21,29,30], or with extracorporeal circulation [24] and stent implantation [31]

Surgical treatment is considered one of the best choices in those severe forms of imperforated CTD necessitating removal of the membrane and atrial defect reparation [12]. Open-heart surgery for heart conditions often involves extracorporeal circulation. During this procedure, a heart–lung machine assumes the functions of the heart and lungs, allowing the heart to be blocked and the surgeon to operate in a bloodless setting. To access the heart, a median sternotomy, where the chest is opened along the midline, is typically performed. Once the heart chamber is exposed, the membrane causing the abnormal division within the heart is surgically removed, restoring normal blood flow [24,37].

In most cases, less invasive approaches can be considered, such as minimally invasive procedures.

Percutaneous balloon dilation is the most common treatment used to correct CTD in symptomatic dogs [7,8,26,27,28], given that is considered to be highly effective, and with rare complications [34].

The balloon technique offers the possibility of introducing a balloon dilatator within the membrane during catheterization, breaking it, and allowing blood to flow normally [7,22]. If necessary, a stent may be used to ensure the obstruction is permanently relieved. This process involves placing a guide wire across the membrane, followed by a balloon catheter. The balloon is inflated to create an opening, and a stent is then deployed to hold the membrane open, ensuring a permanent passage for blood flow [22]. After the procedure, careful monitoring is crucial to detect potential complications like arrhythmias or bleeding. Imaging tests are used in follow-ups to ensure the stent remains in place and that the obstruction has been fully cleared.

As an alternative to the standard balloon dilation, the cutting balloon is a better choice to reduce the risk of restenosis [25]. The focal rupture determined with the cutting balloon approach also reduces the possible onset of accidental dissection, endothelial damage, and the occurrence of arrhythmias [25].

Surgical abstraction of the dividing membrane using cardiopulmonary bypass is also a highly successful curative approach. However, this procedure carries significant risks for small-breed puppies due to their low body weight and limited circulating blood volume, making extracorporeal circulation particularly hazardous [12]. The surgical excision of a right intra-atrial membrane through the placement of a transcatheter stent has also been recently described as another less invasive alternative to open-heart surgery, but it presents important limitations related to the size of blood vessels in low-weight puppies and the need for advanced catheter manipulation skills from the operator [44,45].

Restenosis is a recognized usual complication of CTD treated with cutting ballooning dilation [7,31].

In the case of restenosis, stent implantation and direct surgical resection of the membrane via atriotomy are usually the indicated procedures. In cases of stent implantation, a potential risk could be stent migration, where the stent may move from its intended position.

Another interesting method is described by Akiko Uemura et al. [43], which consists of a hybrid balloon dilation treatment. It is not open-heart surgery and consists of inserting a device directly into the heart with a thoracotomy and, at the same time, the fenestration in the separating membrane is expanded with a balloon. This technique reduces the risk linked to more invasive techniques such as open-heart surgery and removes dependence on the blood vessel diameter.

After the minimally invasive procedures or surgery, dogs require intensive care to monitor recovery and prevent complications. This includes the use of medications to support heart function and constant monitoring through echocardiograms to ensure adequate blood flow.

The benefits of minimally invasive procedures are significant, especially compared to open-heart surgery. These techniques cause less trauma to the body, leading to a quicker recovery for the patient. Additionally, hospital stays tend to be shorter, as patients typically require less time to heal than those who undergo more traditional surgeries. Another advantage is the immediate relief from symptoms that many patients experience once normal blood flow is restored [7,8,26,27,28,34]. However, there are also some risks associated with these procedures. There is a risk of an infection at the site where the stent or catheter is inserted, also, there is the possibility of restenosis, meaning that the treated area could narrow again over time, which may require further medical intervention [7,8,26,27,28,34].

Figure 1 summarizes steps to follow in the management of CTD in dogs.

## 2. Clinical Scenarios and Ethical Considerations

In clinical practice, CTD can occur at different stages and in different contexts; therefore, a unique approach from an ethical and medico-legal viewpoint is not possible. The three different realistic scenarios of CTD occurrence in dogs that follow represent examples for critical considerations from a professional perspective. They were chosen due to their complexity and evident difficulties in applying veterinary ethics theory to practical situations.


*Scenario 1: Adult dog with CTD and associated congenital cardiopathies.*

*A 2-year-old intact male Labrador retriever was referred for inability to exercise and abdominal distension, suggesting a peritoneal effusion. Congestive heart failure on the right side was suspected. Echocardiography showed a perforated CTD associated with pulmonary valve stenosis and tricuspid and mitral valve dysplasia.*


Co-existing CHDs seem to occur frequently in CTD, especially in any breed such as Labrador Retriever, often with aggrieving clinical signs and affecting the prognosis accordingly [7].

Some authors suggest that—when clinical signs are present—surgery is indicated due to a poor response to medical treatment and physical obstruction [7]. However, given the rarity of the clinical picture and the lack of large-scale studies comparing treatment choices, the presence of concomitant CHDs may make the long-term outcome of surgery more difficult.

Here, where one or more courses of action may be appropriate in a given situation as post-operative complications associated with concurrent tricuspid valve dysplasia, veterinary decision making, and ethical reasoning are not addressed in a simple algorithmic way. When CTD is associated with CHDs, various complications may occur after the interventional procedure; therefore, the veterinarian cannot offer a guarantee to owners on the success of treatment. So, the veterinarian should inform the owners about the inherent difficulties involved, the financial effort required, and the possibility that more interventions may be required, highlighting that achieving the aim may be difficult.

In other cases, given the seriousness of the condition and the type of signs present (severe dyspnea, lack of response of pharmacological treatment, or onset of not responsive arrhythmias) and, if no further treatment is possible, it would be advisable to proceed with euthanasia due to concerns about the poor quality of life (QoL). While respecting the wishes of clients, veterinarians must take a more proactive approach when animal welfare issues arise. This involves offering tools to support decision making, such as formal or informal QoL assessments [46,47], to help clients gain a clearer understanding of welfare concerns and to prevent unnecessary pain and suffering. However, this approach aligns with a more utilitarian framework of ethics. Some studies have highlighted that more patient-centered discussions about end-of-life care could be beneficial for owners of dogs with cardiac disease [48].


*Scenario 2: CTD in a juvenile dog (less than 12 months old).*

*A nine-month-old, spayed female, Labrador Retriever, for audible murmur at auscultation associated with jugular distension. Echocardiography showed a perforated membrane that divided the right atrium into two chambers, suggesting CTD. Computed tomography angiography was used to confirm the diagnosis. A ballon membranostomy to reduce the pressure gradient between the two right atrial chambers with consequent resolution of the clinical findings was proposed, but the owner did not consent to the surgical treatment.*


The prognosis for dogs may differ based on the severity of the condition and the response to treatment.

From an ethical perspective, performing a membranostomy to resolve clinical symptoms of CTD in young dogs involves several considerations. First, in terms of animal welfare, the primary ethical obligation is to improve the dog’s QoL by correcting a condition that could lead to long-term suffering or reduced function. If the intervention can resolve the symptoms early, it may prevent future complications and improve the dog’s well-being, which is consistent with the veterinarian’s duty to alleviate suffering. The procedure may resolve the immediate symptoms. However, the long-term welfare of the dog should also be considered. This will include any post-operative care that may be required and to ensure that the intervention does not inadvertently shorten the dog’s life or otherwise reduce the quality of its life. Second, in terms of risk vs. benefit, like any medical intervention, it is necessary to balance the risks of the intervention (e.g., complications of surgery, risks of anesthetic, problems with recovery after surgery) against the potential benefits. Ethically, the veterinarian must ensure that the expected benefits (resolution of symptoms and improvement in QoL) outweigh the risks of harm or pain associated with the surgery.

Finally, in terms of informed consent, the owner needs to be fully informed about the procedure, including the risks, the benefits, the costs, and any possible long-term consequences. Ethical care involves transparency and the provision of sufficient information for the owner to make a decision in the best interests of their pet.

In light of the above, the decision to perform membranostomy must balance immediate relief of suffering with considerations of long-term welfare and informed consent.

In this scenario, the membranostomy intervention could lead to the resolution of clinical symptoms. In fact, balloon membranostomy could be efficacious in decreasing the pressure gradient between the two right atrial chambers with the consequent resolution of the clinical findings. So, the QoL of the dog can be improved, and survival time may be extended.

In this case, the euthanasia of the dog is not justifiable, and the veterinarian has to consider other reasons for the putting down of the animal. Even if animals are unable to participate in the decision-making process, the veterinarian should treat them not as objects (property of the client) but as subjects, patients deserving high-quality medical care [49].


*Scenario 3: CTD in a senior dog (>10 years of age).*

*An 11-year-old, male neutered, mixed breed presented with ascites was referred for the presence of abdominal effusion (ascites) for several weeks. Echocardiography suggested the presence of CTD, not associated with other abnormalities, and confirmed by computed tomography angiography. A few months after the procedure of balloon dilation, ascites reappeared.*


Dogs with CTD may not show symptoms of cardiac congestion until later in life, raising important veterinary considerations. Usually, given that CTD is a congenital defect, the symptoms appear at an early age, although it is reported in cases of adult and old dogs [7,8,9,10,11,12,13,14,21,22,23,24,25,26,27,28,29,30,31]. Why a few dogs remain asymptomatic for years, while others develop the clinical signs early, is unknown [7]. Similarly, it is also reported in people [5,6], where it is suggested that a mechanical stress effect or an age-related remodeling may induce progressive fibrosis and calcification of the orifice.

Even in the absence of overt clinical signs, early detection and proactive management of such conditions is essential to ensure animal welfare. Ethical care requires veterinarians to prioritize QoL through regular monitoring and consideration of interventions that may reduce future complications. This means that veterinarians have an ethical responsibility to focus on the overall well-being and comfort of the animal, rather than merely treating or delaying the progression of disease. In the case of CTD, this could involve ensuring that the animal lives a pain-free, comfortable life through appropriate management, even in the absence of immediate clinical signs.

For example, prioritizing QoL might involve the following:*(i)* Regular monitoring: Regular check-ups to detect early signs of distress or changes in heart health that could prevent suffering later on;*(ii)* Preventive care: Offering medical or lifestyle interventions, such as special diets, medications, or controlled exercise, to prevent the condition from worsening;*(iii)* Owner education: Educating the owner on how to manage the pet’s condition to improve comfort and prevent complications;*(iv)* Decision-making support: In cases where the condition may significantly affect the animal’s future well-being, veterinarians must assist owners in making informed, compassionate decisions about treatment options, including palliative care or euthanasia, when suffering becomes unmanageable.

Prioritizing QoL therefore means ensuring that every medical decision is made with the aim of minimizing pain and maximizing comfort for the animal throughout its lifetime.

Figure 2 shows a stepwise decision tree for an approach to CTD in dogs.

## 3. Discussion

In veterinary medicine, it is essential to consider the severity of the patient’s disease to propose the most appropriate treatment, including euthanasia. Whatever the disease, it is always difficult for veterinarians to understand when the medical treatment does not make a significant difference to the patient’s condition. Specifically, for CTD, they have to rate the patient’s condition following some steps to suggest the most correct treatment. The first step is to check the symptoms present. Dogs affected by CTD often do not need minimally invasive techniques or surgery to solve the defect, because their life can be sustained by specific comportments, like maintaining a quiet lifestyle (especially in adult or old patients) combined with drug treatment support [7,8,9,10,11,12,13,14,21,22,23,24,25,26,27,28,29,30,31].

The second step is to perform all complete diagnostic activities to rate CTD and understand if there are concomitant congenital and/or consequential diseases. The third step is to value what technique (minimally invasive or surgery) is appropriate for the single patient, considering mortality, post-treatment patient lifestyle, and restarting symptom possibilities due to side effects.

Based on the literature, CTD may appear as a silent bystander or a potential time bomb, given that the occurrence of clinical signs is strongly related to the location and the integrity of the membrane, and any concurrent associated CHDs. Asymptomatic and symptomatic young patients with perforate CTD and no concurrent cardiac disease have a very high possibility of solving the disease by minimally invasive procedures or surgery with rare recurrent symptoms after a six-month to two-year follow-up period [7,8,9,10,11,12,13,14]. Asymptomatic and symptomatic young patients with perforate CTD and concurrent cardiac abnormalities have a mid–high possibility of solving the disease by minimally invasive procedures or surgery with rare recurrent symptoms after a six-month to two-year follow-up period [7]. Studies about symptomatic young patients with imperforate CTD are more conflicting; in fact, there are documented cases of intraoperative deaths [7,8], but also cases where the disease was cured using one or more procedures [7,8,9,10,11,12,13,14,21,22,23,24,25,26,27,28,29,30,31]. Specifically, when young symptomatic patients have imperforate CTD mixed with congenital and/or consequential heart disease, intraoperative survival rates seriously decrease, and life expectation post-treatment is short [7,8,22,25,26,34]. Therefore, when CTD is imperforate, mixed with concurrent congenital and/or consequential heart disease, and patients present severe symptoms like ascites and/or pulmonary infiltrates that are no longer reduced by drug therapy, euthanasia may appear to be the best choice, even if there is no specific literature about this. The prognosis for dogs can vary depending on how severe the condition is and how well they respond to treatment. With proper medical management, their quality of life can be improved, and survival time may be extended [7,8,35,36,43].

Another relevant aspect regards the cost of treatment. Indeed, very often owners cannot sustain too-expansive procedures like minimally invasive and surgical treatments, and, specifically, for that disease, treatment and hospitalization costs could be very high. Referring to CTD, it is common for owners to choose euthanasia in cases where the pet needs a specific diagnostic and care treatment that has too high a cost. As veterinarian doctors, our role should be to safeguard patient health and welfare, so euthanasia should be proposed only in cases where it is extremely necessary and where the patient’s condition offers that choice as one of the reported options. Moreover, it would be helpful to incentivize owners to establish veterinary insurance that could cover the total or a part of the medical cost in cases of diseases that require high diagnostic and care treatment costs such as CTD.

## 4. Conclusions

CTD is a rare congenital cardiac anomaly with various clinical signs of right heart failure and is often associated with other cardiovascular abnormalities. Though it is occasionally described, its incidence is unknown. Therefore, its diagnosis and complete categorization of the malformation in isolated or associated form necessitates a meticulous investigation and a multimodal approach, including also an ethical evaluation.

## Figures and Tables

**Figure 1 vetsci-12-00175-f001:**
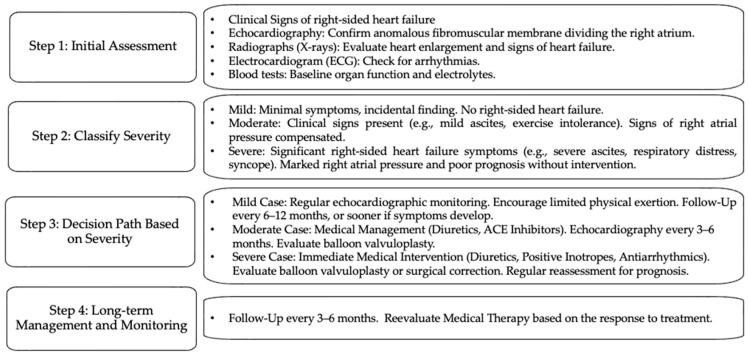
Decision tree for managing CTD in dogs based on disease severity and clinical presentation.

**Figure 2 vetsci-12-00175-f002:**
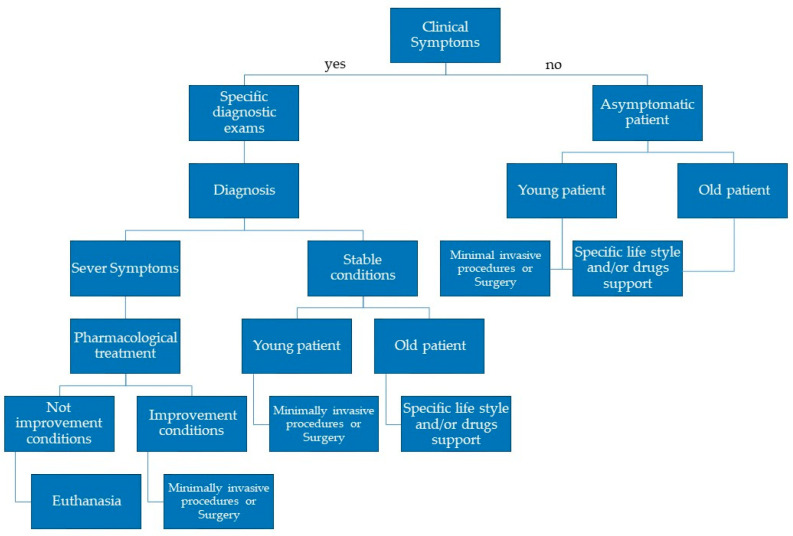
A stepwise decision tree for approaching CTD in dogs.

**Table 1 vetsci-12-00175-t001:** Clinical signs reported in canine CTD [6].

Clinical Signs	
Cardiac and Respiratory Signs	Other Signs
Cardiac murmur Respiratory distress Pleural effusion Cyanosis	Ascites Hepatomegaly Exercise intolerance Stunted growth Vomiting

CTD; *Cor triatriatum dexter*.

**Table 2 vetsci-12-00175-t002:** Concurrent CHDs associated with CTD in dogs.

Valves Abnormalities	Septal Defects	Vascular and Pericardial Abnormalities
Tricuspid valve dysplasia or aplasia Pulmonic stenosis Subaortic stenosis Ebstein’s anomaly	Atrial septal defect Ventricular septal defect *Patent foramen ovale*	Pericardial agenesis Persistent left cranial vena cava Patent Ductus Arteriosus

CHDs; Congenital heart diseases.

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
