# Peer review of "Cor triatriatum dexter in Dogs: A Silent Bystander or a Potential Time Bomb? A Focused Literature Review from a Professional Perspective"

_vetsci, 2025, doi:10.3390/vetsci12020175_

Round 1
Reviewer 1 Report
Comments and Suggestions for Authors
Reviewers comments
Cor triatriatum dexter in dog: a silent bystander or a potential 2 time bomb? A focused literature review from a professional 3 perspective
In general:
CTD is realy interresting congenital anomaly in dogs and trying to make an approach with possibilities on one side and ethical problems on the other side is interresting.
From my point of view the diagnostic possibilities should be shown in more detail, because if you want to discuss like you do later on with the three cases a more detailed explanation of the patients and the type of CTD would be necessary.
You have have done a good search of the complete veterinary literature of the cases, small studies and one bigger study with 17 dogs. But in the text you don’t realy use all these informations and stay at one or two statements. Please give a complete overview, especially of the procedures which are possible and done, please also amke realy differentiation between surgery and catheter intervention. Sometimes I have the impression that your approach is a little bit negativ about performing catheter intervention and this is realy a nice and good possibility to treat these patients. Please talk a little bit more about the different types( perorate, imperforate, additional diseases…). Your approach about medical treatment is very theoretical, because if there is a congestion it is like in many other diseases. Please think about some medications if they realy have a chance to change something in these patients like a pimobendan.
It would be fantastic if you could adopt this review, but currently there is some work to do. Please look on the detailed comments.
Please check the complete paper, because there are some english spelling problems.
Simple Summary
Line 14: please insert that it is the right atrium
Line 15: If you talk here about heart failure cyanosis or sudden death is from the current literature not so common, escecially if it is only a CTD. First sign of heart failure is then effusion. Cyanosis is more often if there are additional defects. Please rewrite this scentence
Abstract
Line 23: like above please insert right atrium
Line 25: it is the same like above
One comment: it would be essential to describe here in the abstract that the CTD is an anomaly which is very often associated with other congenital anomalies.
Introduction:
Line 38: I am sorry but it don’t understand that secntence: „
This review explores the pathophysiology, diagnostic approaches, and treatment modalities for major 39 CHDs“
è The theme of this publication is CTD and not major CHDs
è Please rewrite this.
Line 40: These two references are only tectbooks ? recent scientific advancements are more specific publications. Please use maybee additional references for this comment.
Line 61: You describe all the possible additional congenital cardiac anomalies, at the end of this scetence references would be fine.
Line 62: Is this part necessary, because the title is CTD?
Line 65: please check reference, because 5 is in dogs and only for CTD and not CTS
Clinical signs
Line 76: Please check again the references, 5 is veterinary literature
Line 89: This table represents not the facts you describe before and is in this way not acceptable
What is the intention with arrhythmias, because mostly sinus rhythm is present. Please check this complete part and rewrite the table if it is really necessary
Table 2:
In the first column: if you arrange it to the prevalence you have to change the two stenosis types
In the second column: It is mostly called patent foramen ovale
In the third column: You have to delete the Ebsteins anomaly at this position because it is a special type of tricuspid valve dysplasia
Diagnosis
Line 105: Please do not use the term „ventricle“ for the two parts of the right atrium, you can use compartment….
Line 108: The complete echo part is nice, but for a review article detailed informations are issing. Please think about going more into detail, maybe some nice echo pictures.
Line 110: You describe that there are one or more orifices in the abnormal membrane, but you do not talk about the patients with an imperforate type, please insert this
Line 117: This is very difficult because a hypertrophy in the atrium is not defined in the current literature, please clarify this
Pharmacological and Surgical Treatment
Line 141: Please correct that the only therapy is surgical, because there a lot of catheter based options
Line 142: I am sorry but this is defenetively incorrect! A diuretic is furosemide, maybe you can say also Spironolacton, but Benazepril is a ACE-inhibitior and Pimobendan ist a inodilator. Please correct this
Line 144: There is no congestion in the lungs, it is no left heart backward failure. You can find pleural effusion, please correct this
Line 150: This a theoretic concept, if you use high dosages of furosemide only giving Spironolacton will not be enough
Line 164: Please specify this scentence because you should differentiate surgery and catheter intervention/minimal invasive procedure
Line 167: You describe that surgery is the first option, later on you describe that there are less invasive procedures. Normally the first approach is catheter intervention because it is easier and less invasive, please clarify this
Line 191: It is correct that a stent offers some potential risks, but currently stent is not the initial procedure, because mostly only ballooning the valve with or without cutting balloon is enough. Stent is only necessary if there is a re-stenosis or the membrane is to stiff. Please cleary this in this paragraph.
Line 197: In the Figure 1
à in step 3 you describe giving positive inotropes, what is the reason, what do you expect that the positive inotrope should do? You have described above the way they work, please clarify
Clinical scenarios and ethical considerations
Line 221:
This is really very theoretical approach.
CTD, especially if combined with other congenital cardiac diseases is a complex cardiac malformation and you will not have a garante if you offer the owners a catheter intervention, but anesthesia is not the big problem (is published) and directly to talk about potential problems with a stent is not a good work. It is true that it could be difficult and maybe you need more than one inteervention, but the aim you could achieve is realy great. One problem beside the risks of such a procedure is the money and that is a big point the owners have to know. It is true if the dog is in congestion and there is no perspective and oral medication doesn’t work yes you have to talk about euthanasia. Please think about these aspects.
Line 312:
In you flow chart you directly say if a dog with severe symptoms which not reat on a pharmacological therapy you have to do euthanasia. I don’t agree. First thing why should a dog with congestion due to CTD and maybe other CHD not respond to i.v. medication or puncture of fluid accululation. And sometimes a little bit better is enough to do intervention and achieve a good outcome. Please think about this.
Is there another disease and there is no option to make it better okay, but that is not in this flow chart
Discussion
Please check also in this part the term surgery, and clarify if you realy mean surgery or is it more catheter intervention.
Line 329: You use for your statement for this disease group only a human paper review article, that is not corrrect, please use the veterinary literatur for this, it is not easy I know.
Line 334: This literatur you citate (number 6) is incorrect for this setting, because it is an imperforate type
The next part about imperforate CTD is not correct in this way, please change it an use veterinary literature, there are some cases with imperforate types of CTD, human data is not of use in dogs because there are cases with realy old people.
Line 346
I am sorry but this is no realy discussion. You use one or two papers.
The next thing is you explain again surgery is the best approach. Not at all, because it is realy complicated, you need much more material and good specialists for this procedure.
There are realy many cases and a review with 17 dogs in which mostly if neccessary a catheter based approach was used and in dogs with a perforate type this a procedure which is not that complicated. Yes possible problems can happen, but normally also in small breed dogs with only some kg body weight it works fine.
A personal question, how many procedures, catheter or surgery have you seen or done?
Comments on the Quality of English Language
Reviewers comments
Cor triatriatum dexter in dog: a silent bystander or a potential 2 time bomb? A focused literature review from a professional 3 perspective
In general:
CTD is realy interresting congenital anomaly in dogs and trying to make an approach with possibilities on one side and ethical problems on the other side is interresting.
From my point of view the diagnostic possibilities should be shown in more detail, because if you want to discuss like you do later on with the three cases a more detailed explanation of the patients and the type of CTD would be necessary.
You have have done a good search of the complete veterinary literature of the cases, small studies and one bigger study with 17 dogs. But in the text you don’t realy use all these informations and stay at one or two statements. Please give a complete overview, especially of the procedures which are possible and done, please also amke realy differentiation between surgery and catheter intervention. Sometimes I have the impression that your approach is a little bit negativ about performing catheter intervention and this is realy a nice and good possibility to treat these patients. Please talk a little bit more about the different types( perorate, imperforate, additional diseases…). Your approach about medical treatment is very theoretical, because if there is a congestion it is like in many other diseases. Please think about some medications if they realy have a chance to change something in these patients like a pimobendan.
It would be fantastic if you could adopt this review, but currently there is some work to do. Please look on the detailed comments.
Please check the complete paper, because there are some english spelling problems.
Simple Summary
Line 14: please insert that it is the right atrium
Line 15: If you talk here about heart failure cyanosis or sudden death is from the current literature not so common, escecially if it is only a CTD. First sign of heart failure is then effusion. Cyanosis is more often if there are additional defects. Please rewrite this scentence
Abstract
Line 23: like above please insert right atrium
Line 25: it is the same like above
One comment: it would be essential to describe here in the abstract that the CTD is an anomaly which is very often associated with other congenital anomalies.
Introduction:
Line 38: I am sorry but it don’t understand that secntence: „
This review explores the pathophysiology, diagnostic approaches, and treatment modalities for major 39 CHDs“
è The theme of this publication is CTD and not major CHDs
è Please rewrite this.
Line 40: These two references are only tectbooks ? recent scientific advancements are more specific publications. Please use maybee additional references for this comment.
Line 61: You describe all the possible additional congenital cardiac anomalies, at the end of this scetence references would be fine.
Line 62: Is this part necessary, because the title is CTD?
Line 65: please check reference, because 5 is in dogs and only for CTD and not CTS
Clinical signs
Line 76: Please check again the references, 5 is veterinary literature
Line 89: This table represents not the facts you describe before and is in this way not acceptable
What is the intention with arrhythmias, because mostly sinus rhythm is present. Please check this complete part and rewrite the table if it is really necessary
Table 2:
In the first column: if you arrange it to the prevalence you have to change the two stenosis types
In the second column: It is mostly called patent foramen ovale
In the third column: You have to delete the Ebsteins anomaly at this position because it is a special type of tricuspid valve dysplasia
Diagnosis
Line 105: Please do not use the term „ventricle“ for the two parts of the right atrium, you can use compartment….
Line 108: The complete echo part is nice, but for a review article detailed informations are issing. Please think about going more into detail, maybe some nice echo pictures.
Line 110: You describe that there are one or more orifices in the abnormal membrane, but you do not talk about the patients with an imperforate type, please insert this
Line 117: This is very difficult because a hypertrophy in the atrium is not defined in the current literature, please clarify this
Pharmacological and Surgical Treatment
Line 141: Please correct that the only therapy is surgical, because there a lot of catheter based options
Line 142: I am sorry but this is defenetively incorrect! A diuretic is furosemide, maybe you can say also Spironolacton, but Benazepril is a ACE-inhibitior and Pimobendan ist a inodilator. Please correct this
Line 144: There is no congestion in the lungs, it is no left heart backward failure. You can find pleural effusion, please correct this
Line 150: This a theoretic concept, if you use high dosages of furosemide only giving Spironolacton will not be enough
Line 164: Please specify this scentence because you should differentiate surgery and catheter intervention/minimal invasive procedure
Line 167: You describe that surgery is the first option, later on you describe that there are less invasive procedures. Normally the first approach is catheter intervention because it is easier and less invasive, please clarify this
Line 191: It is correct that a stent offers some potential risks, but currently stent is not the initial procedure, because mostly only ballooning the valve with or without cutting balloon is enough. Stent is only necessary if there is a re-stenosis or the membrane is to stiff. Please cleary this in this paragraph.
Line 197: In the Figure 1
à in step 3 you describe giving positive inotropes, what is the reason, what do you expect that the positive inotrope should do? You have described above the way they work, please clarify
Clinical scenarios and ethical considerations
Line 221:
This is really very theoretical approach.
CTD, especially if combined with other congenital cardiac diseases is a complex cardiac malformation and you will not have a garante if you offer the owners a catheter intervention, but anesthesia is not the big problem (is published) and directly to talk about potential problems with a stent is not a good work. It is true that it could be difficult and maybe you need more than one inteervention, but the aim you could achieve is realy great. One problem beside the risks of such a procedure is the money and that is a big point the owners have to know. It is true if the dog is in congestion and there is no perspective and oral medication doesn’t work yes you have to talk about euthanasia. Please think about these aspects.
Line 312:
In you flow chart you directly say if a dog with severe symptoms which not reat on a pharmacological therapy you have to do euthanasia. I don’t agree. First thing why should a dog with congestion due to CTD and maybe other CHD not respond to i.v. medication or puncture of fluid accululation. And sometimes a little bit better is enough to do intervention and achieve a good outcome. Please think about this.
Is there another disease and there is no option to make it better okay, but that is not in this flow chart
Discussion
Please check also in this part the term surgery, and clarify if you realy mean surgery or is it more catheter intervention.
Line 329: You use for your statement for this disease group only a human paper review article, that is not corrrect, please use the veterinary literatur for this, it is not easy I know.
Line 334: This literatur you citate (number 6) is incorrect for this setting, because it is an imperforate type
The next part about imperforate CTD is not correct in this way, please change it an use veterinary literature, there are some cases with imperforate types of CTD, human data is not of use in dogs because there are cases with realy old people.
Line 346
I am sorry but this is no realy discussion. You use one or two papers.
The next thing is you explain again surgery is the best approach. Not at all, because it is realy complicated, you need much more material and good specialists for this procedure.
There are realy many cases and a review with 17 dogs in which mostly if neccessary a catheter based approach was used and in dogs with a perforate type this a procedure which is not that complicated. Yes possible problems can happen, but normally also in small breed dogs with only some kg body weight it works fine.
A personal question, how many procedures, catheter or surgery have you seen or done?
Author Response
Dear Reviewer,
Thank you very much for your time and all your comments.
We thank you for your precise and thoughtful comments and constructive criticism, which has led to a better manuscript.
We revised the manuscript concerning the suggestions and more detailed answers are given below.
The changes made in the manuscript to address comments are written in red.
- In general:
CTD is realy interresting congenital anomaly in dogs and trying to make an approach with possibilities on one side and ethical problems on the other side is interresting.
From my point of view the diagnostic possibilities should be shown in more detail, because if you want to discuss like you do later on with the three cases a more detailed explanation of the patients and the type of CTD would be necessary.
You have have done a good search of the complete veterinary literature of the cases, small studies and one bigger study with 17 dogs. But in the text you don’t realy use all these informations and stay at one or two statements. Please give a complete overview, especially of the procedures which are possible and done, please also amke realy differentiation between surgery and catheter intervention. Sometimes I have the impression that your approach is a little bit negativ about performing catheter intervention and this is realy a nice and good possibility to treat these patients. Please talk a little bit more about the different types( perorate, imperforate, additional diseases…). Your approach about medical treatment is very theoretical, because if there is a congestion it is like in many other diseases. Please think about some medications if they realy have a chance to change something in these patients like a pimobendan.
It would be fantastic if you could adopt this review, but currently there is some work to do. Please look on the detailed comments.
Please check the complete paper, because there are some english spelling problems.
- The diagnostic possibilities are included more detailly, and a complete overview, of the interventional procedures have been added (lines 179-199; 206-209; 214-231). Different types of CTD have been described (lines 55-57). The paragraph regarding the medical treatment has been modified (lines 155-165). The English language has been reviewed.
Simple Summary
- Line 14: please insert that it is the right atrium
- Done
- Line 15: If you talk here about heart failure cyanosis or sudden death is from the current literature not so common, escecially if it is only a CTD. First sign of heart failure is then effusion. Cyanosis is more often if there are additional defects. Please rewrite this scentence
- The sentence has been rewritten as suggested
Abstract
- Line 23: like above please insert right atrium
- Done
- Line 25: it is the same like above
One comment: it would be essential to describe here in the abstract that the CTD is an anomaly which is very often associated with other congenital anomalies.
Introduction:
- Line 38: I am sorry but it don’t understand that secntence: „
This review explores the pathophysiology, diagnostic approaches, and treatment modalities for major 39 CHDs“.The theme of this publication is CTD and not major CHDs. Please rewrite this.
- The sentence has been deleted.
- Line 40: These two references are only tectbooks ? recent scientific advancements are more specific publications. Please use maybee additional references for this comment.
- Done
- Line 61: You describe all the possible additional congenital cardiac anomalies, at the end of this scetence references would be fine.
- Done
- Line 62: Is this part necessary, because the title is CTD?
- Done
- Line 65: please check reference, because 5 is in dogs and only for CTD and not CTS
Clinical signs
- Done
- Line 76: Please check again the references, 5 is veterinary literature
- Done
- Line 89: This table represents not the facts you describe before and is in this way not acceptable
What is the intention with arrhythmias, because mostly sinus rhythm is present. Please check this complete part and rewrite the table if it is really necessary
Table 2:
In the first column: if you arrange it to the prevalence you have to change the two stenosis types
In the second column: It is mostly called patent foramen ovale
In the third column: You have to delete the Ebsteins anomaly at this position because it is a special type of tricuspid valve dysplasia
- The table has been modified as suggested.
Diagnosis
- Line 105: Please do not use the term „ventricle“ for the two parts of the right atrium, you can use compartment….
- Done
- Line 108: The complete echo part is nice, but for a review article detailed informations are issing. Please think about going more into detail, maybe some nice echo pictures.
- The echocardiography examination has been improved with other detailed information.
- Line 110: You describe that there are one or more orifices in the abnormal membrane, but you do not talk about the patients with an imperforate type, please insert this
- Done
- Line 117: This is very difficult because a hypertrophy in the atrium is not defined in the current literature, please clarify this
- The sentence has been modified.
Pharmacological and Surgical Treatment
- Line 141: Please correct that the only therapy is surgical, because there a lot of catheter based options
- Line 142: I am sorry but this is defenetively incorrect! A diuretic is furosemide, maybe you can say also Spironolacton, but Benazepril is a ACE-inhibitior and Pimobendan ist a inodilator. Please correct this
- Line 144: There is no congestion in the lungs, it is no left heart backward failure. You can find pleural effusion, please correct this
A.Done.
- Line 150: This a theoretic concept, if you use high dosages of furosemide only giving Spironolacton will not be enough
- Line 164: Please specify this scentence because you should differentiate surgery and catheter intervention/minimal invasive procedure
- Line 167: You describe that surgery is the first option, later on you describe that there are less invasive procedures. Normally the first approach is catheter intervention because it is easier and less invasive, please clarify this
- Line 191: It is correct that a stent offers some potential risks, but currently stent is not the initial procedure, because mostly only ballooning the valve with or without cutting balloon is enough. Stent is only necessary if there is a re-stenosis or the membrane is to stiff. Please cleary this in this paragraph.
- A. The paragraph has been modified.
- Line 197: In the Figure 1
à in step 3 you describe giving positive inotropes, what is the reason, what do you expect that the positive inotrope should do? You have described above the way they work, please clarify - The paragraph has been modified.
Clinical scenarios and ethical considerations
- Line 221:This is really very theoretical approach.
CTD, especially if combined with other congenital cardiac diseases is a complex cardiac malformation and you will not have a garante if you offer the owners a catheter intervention, but anesthesia is not the big problem (is published) and directly to talk about potential problems with a stent is not a good work. It is true that it could be difficult and maybe you need more than one inteervention, but the aim you could achieve is realy great. One problem beside the risks of such a procedure is the money and that is a big point the owners have to know. It is true if the dog is in congestion and there is no perspective and oral medication doesn’t work yes you have to talk about euthanasia. Please think about these aspects.
- The paragraph has been modified as suggested.
Line 312:
- In you flow chart you directly say if a dog with severe symptoms which not reat on a pharmacological therapy you have to do euthanasia. I don’t agree. First thing why should a dog with congestion due to CTD and maybe other CHD not respond to i.v. medication or puncture of fluid accululation. And sometimes a little bit better is enough to do intervention and achieve a good outcome. Please think about this.
Is there another disease and there is no option to make it better okay, but that is not in this flow chart
- Flow chart has been rewritten as suggested, including the differentiation between minimal invasive procedures and surgery
Discussion
Please check also in this part the term surgery, and clarify if you realy mean surgery or is it more catheter intervention.
Line 329: You use for your statement for this disease group only a human paper review article, that is not corrrect, please use the veterinary literatur for this, it is not easy I know.
Line 334: This literatur you citate (number 6) is incorrect for this setting, because it is an imperforate type
The next part about imperforate CTD is not correct in this way, please change it an use veterinary literature, there are some cases with imperforate types of CTD, human data is not of use in dogs because there are cases with realy old people.
- Line 346\
I am sorry but this is no realy discussion. You use one or two papers.
The next thing is you explain again surgery is the best approach. Not at all, because it is realy complicated, you need much more material and good specialists for this procedure.
There are realy many cases and a review with 17 dogs in which mostly if neccessary a catheter based approach was used and in dogs with a perforate type this a procedure which is not that complicated. Yes possible problems can happen, but normally also in small breed dogs with only some kg body weight it works fine.
A personal question, how many procedures, catheter or surgery have you seen or done?
A: Thank you for your suggestions. A substantial improvement of this paragraph has been performed including more detailed information. For a better understanding for the reader this paragraph has been moved above (Pharmacological and Surgical Treatment).
Best Regards
Prof. Michela

Reviewer 2 Report
Comments and Suggestions for Authors
The title of the article is provocative but inadequate to the content.
The defect as such, Cor triatriatum dexter, is rare in dogs and few cases have been described (the largest group is 17 dogs). Therefore, no treatment algorithms have been developed so far.
In my opinion, dividing patients into young and old without defining ‘young’ and ‘old’ makes the algorithm poorly useful. An important aspect of surgical treatment is the experience of the team, which the authors do not discuss in their article. Among the treatment methods, the authors did not include hybrid treatment (Uemura A, Yoshida T, Matsuura K, Yilmaz Z, Tanaka R. Hybrid balloon dilation treatment for cor triatriatum dexter in a small breed puppy. J Vet Sci. 2019 Aug;20(5):e49. https://doi.org/10.4142/jvs.2019.20.e49)
Author Response
Dear Reviewer,
Thank you very much for your time and all your comments.
We thank you for your precise and thoughtful comments and constructive criticism, which has led to a better manuscript.
We revised the manuscript concerning the suggestions and more detailed answers are given below.
The changes made in the manuscript to address comments are written in red.
- The title of the article is provocative but inadequate to the content.
The defect as such, Cor triatriatum dexter, is rare in dogs and few cases have been described (the largest group is 17 dogs). Therefore, no treatment algorithms have been developed so far.
- The title has been connected at the discussion.
- In my opinion, dividing patients into young and old without defining ‘young’ and ‘old’ makes the algorithm poorly useful. An important aspect of surgical treatment is the experience of the team, which the authors do not discuss in their article.
- Done.
- Among the treatment methods, the authors did not include hybrid treatment (Uemura A, Yoshida T, Matsuura K, Yilmaz Z, Tanaka R. Hybrid balloon dilation treatment for cor triatriatum dexter in a small breed puppy. J Vet Sci. 2019 Aug;20(5):e49. https://doi.org/10.4142/jvs.2019.20.e49)
- The hybrid treatment method has been included as you suggested.
Best Regards
Prof. Michela Pugliese

Reviewer 3 Report
Comments and Suggestions for Authors
This review is well organized and clear. CTD is a rare congenital cardiac anomaly that can have various clinical signs of right heart failure, often associated with other cardiovascular anomalies. The incidence of the pathology in veterinary medicine is unknown, and a systematic approach can be useful to make a diagnosis and a complete categorization of the malformation in isolated or associated form. The clinical and diagnostic approach aim to stage the degree of the pathology and to propose the most appropriate treatment to the patient also based on the availability of the owner.
Line 135 "In a few cases...amplitude (>0.5 mV; normal ≤ 0.4mV), I would recommend removing this sentence or adding some supporting bibliography.
The bibliographic sources are numerous, but often taken from textbooks and some very dated, I would recommend reducing the bibliographic sources and leaving only the significant ones.
-pulmonale, characterized by a large amplitude (> 1)
Author Response
Dear Reviewer,
Thank you very much for your time and all your comments.
We thank you for your precise and thoughtful comments and constructive criticism, which has led to a better manuscript.
We revised the manuscript concerning the suggestions and more detailed answers are given below.
The changes made in the manuscript to address comments are written in red.
- This review is well organized and clear. CTD is a rare congenital cardiac anomaly that can have various clinical signs of right heart failure, often associated with other cardiovascular anomalies. The incidence of the pathology in veterinary medicine is unknown, and a systematic approach can be useful to make a diagnosis and a complete categorization of the malformation in isolated or associated form. The clinical and diagnostic approach aim to stage the degree of the pathology and to propose the most appropriate treatment to the patient also based on the availability of the owner.
- Line 135 "In a few cases...amplitude (>0.5 mV; normal ≤ 0.4mV), I would recommend removing this sentence or adding some supporting bibliography.
- The bibliographic sources are numerous, but often taken from textbooks and some very dated, I would recommend reducing the bibliographic sources and leaving only the significant ones.
- The bibliographic sources have been reduced as suggested.
Best Regards
Prof. Michela Pugliese

Round 2
Reviewer 1 Report
Comments and Suggestions for Authors
Reviewers comments
Cor triatriatum dexter in dog: a silent bystander or a potential 2 time bomb? A focused literature review from a professional 3 perspective
In general:
Thanks to the authors for the work they did improving the quality of this paper. Now there are only two small things to do. CTD is a congenital anomaly which is in every patient an individual case, which should be interpretated individually. This paper can now help veterinarians to make there diagnosis and decide which therapy is good.
Simple Summary
Line 15: Congestion in these patients normaly generates an ascites not a plural effusion, please correct this
Abstract
This part is now finr for me, thanks for the corrections
Introduction:
Line 66: You still have this part about the CTS included. Again the question the theme is CTD and not CTS. You write as answer „done“. For me it is not needed and should be deleted. Please think about this.
Clinical signs
This part is now finr for me, thanks for the corrections
Diagnosis
This part is now finr for me, thanks for the corrections
Pharmacological and Surgical Treatment
This part is now finr for me, thanks for the corrections
Clinical scenarios and ethical considerations
This part is now finr for me, thanks for the corrections
Discussion
This part is now finr for me, thanks for the corrections
Author Response
Dear Reviewer,
Thank you very much for your time and all your comments.
We revised the manuscript concerning the suggestions and more detailed answers are given below.
The changes made in the manuscript to address comments are written in red.
Reviewers comments
Cor triatriatum dexter in dog: a silent bystander or a potential 2 time bomb? A focused literature review from a professional 3 perspective
In general:
- Thanks to the authors for the work they did improving the quality of this paper. Now there are only two small things to do. CTD is a congenital anomaly which is in every patient an individual case, which should be interpretated individually. This paper can now help veterinarians to make there diagnosis and decide which therapy is good.
- Thank you for your comments.
Simple Summary
- Line 15: Congestion in these patients normaly generates an ascites not a plural effusion, please correct this
- Done.
Abstract
- This part is now finr for me, thanks for the corrections.
- Thank you for your comments.
Introduction:
- Line 66: You still have this part about the CTS included. Again the question the theme is CTD and not CTS. You write as answer „done“. For me it is not needed and should be deleted. Please think about this.
- Done.
Clinical signs
- This part is now finr for me, thanks for the corrections
- Thank you.
Diagnosis
- This part is now finr for me, thanks for the corrections.
- Thank you.
Pharmacological and Surgical Treatment
- This part is now finr for me, thanks for the corrections.
- Thank you.
Clinical scenarios and ethical considerations
- This part is now finr for me, thanks for the corrections
- Thank you.
Discussion
- This part is now finr for me, thanks for the corrections.
- Thank you.
Best Regards
Prof. Michela Pugliese
